# Updates on the Applications of Spectral Computed Tomography for Musculoskeletal Imaging

**DOI:** 10.3390/diagnostics14070732

**Published:** 2024-03-29

**Authors:** Liesl S. Eibschutz, George Matcuk, Michael Kuo-Jiun Chiu, Max Yang Lu, Ali Gholamrezanezhad

**Affiliations:** 1Department of Radiology, Keck School of Medicine, University of Southern California (USC), Los Angeles, CA 90007, USAmichael.chiu@med.usc.edu (M.K.-J.C.); maxl0147@outlook.com (M.Y.L.); 2Department of Radiology, Cedars-Sinai Medical Center, Los Angeles, CA 90048, USA

**Keywords:** spectral computed tomography, dual-energy computed tomography, virtual non-calcium reconstruction algorithms, musculoskeletal imaging, bone-marrow edema

## Abstract

Spectral CT represents a novel imaging approach that can noninvasively visualize, quantify, and characterize many musculoskeletal pathologies. This modality has revolutionized the field of radiology by capturing CT attenuation data across multiple energy levels and offering superior tissue characterization while potentially minimizing radiation exposure compared to traditional enhanced CT scans. Despite MRI being the preferred imaging method for many musculoskeletal conditions, it is not viable for some patients. Moreover, this technique is time-consuming, costly, and has limited availability in many healthcare settings. Thus, spectral CT has a considerable role in improving the diagnosis, characterization, and treatment of gout, inflammatory arthropathies, degenerative disc disease, osteoporosis, occult fractures, malignancies, ligamentous injuries, and other bone-marrow pathologies. This comprehensive review will delve into the diverse capabilities of dual-energy CT, a subset of spectral CT, in addressing these musculoskeletal conditions and explore potential future avenues for its integration into clinical practice.

## 1. Introduction

For decades, clinicians have strived to noninvasively detect and characterize musculoskeletal conditions ranging from bone and joint pathologies to primary bone malignancies and metastases. The advent of dual-energy computed tomography (DECT), a subset of spectral computed tomography (CT) imaging, has ushered in a new era in radiology, allowing for material differentiation, unparalleled tissue characterization, robust quantification, and a reduction in iodine dosage [1]. This technique’s success involves its ability to acquire CT attenuation data at two distinct energy levels simultaneously. While magnetic resonance imaging (MRI) remains the gold standard for various musculoskeletal pathologies, its practical application often encounters barriers, especially in the emergent setting and with claustrophobic patients or those with non-compatible hardware. This review aims to explore spectral CT’s established role in diagnosing gout and delve into its diverse applications within musculoskeletal imaging, such as bone-marrow edema as a marker of non-displaced or subtle fractures, degenerative disc disease, ligamentous and cartilaginous injury, malignancies, metal-artifact reduction, and inflammatory arthropathies. By identifying the role of DECT in diagnosing, grading, and treating various musculoskeletal pathologies, we will ultimately elucidate its pivotal role in advancing patient care.

## 2. Methods

We conducted a series of literature searches from November 2023 to February 2024 within the PubMed database using combinations of the following terms: “CT”; “computed tomography”; “dual energy”; “dual-energy”; “dual-source”; “dual-layer”; “dual layer”; spectral; “DECT”; “musculoskeletal imaging”; “gout”; “calcium pyrophosphate”; “collagen mapping”; “Arthritis, “inflammatory arthropathies”; Rheumatoid”[Mesh]; “Arthritis, Psoriatic”[Mesh]; “bone marrow edema”; “edema”; “degenerative disc disease”; “osteoporosis”; “multiple myeloma”; “primary malignant bone tumors”; “Neoplasms, Bone Tissue”[Mesh]; “Bone Neoplasms”[Mesh]; “metal artifact reduction”; and “virtual non-calcium reconstruction algorithms”. The search results were not restricted by year, language, or article type, though articles were screened to only include English-language primary research articles and review articles (including systematic reviews and meta-analyses). The references contained within relevant primary research articles and review articles were also screened for inclusion.

## 3. Overview of Spectral CT

Spectral CT encompasses various CT techniques, including dual-energy and more recently, photon-counting CT (Table 1). Spectral CT operates by acquiring data at multiple energy levels sequentially or simultaneously, providing enhanced tissue characterization and material differentiation [1]. DECT was one of the first implementations of spectral CT and can be divided into dual source, rapid kilovolt peak switching, split-filter twin beam, dual spin, and dual layer. Dual-source CT employs two separate X-ray sources at differing energy levels, rapid kilovolt peak switching alternates between high and low energy levels during the scan, and dual-spin and split-filter twin beam utilize a single X-ray source with either a spinning filter or a split beam to produce differing energy spectra [2]. One advantage of the dual-spin method is that it can also be utilized with any CT scanner to obtain CT projections at more than one energy level [3]. While the prior DECT variants discussed are all source-based modalities, dual-layer and photon-counting CT are detector-based technologies [4]. Dual-layer CT differs from other DECT techniques, as it uses one X-ray source but has two layered detectors that collect high and low energy spectra, respectively, thus helping reduce artifact and spectral separation [3]. Each of these DECT subsets has advantages and limitations with respect to radiation dose, spectral separation, and imaging speed; thus, these modalities have no one-size-fits-all clinical application [5]. Photon-counting CT is another spectral CT modality that does not fall under the umbrella of DECT. This comprises the newest technology and directly counts individual photons at varying energy levels, allowing for improved contrast, spatial resolution, and lower radiation doses [3,6]. Ultimately, spectral CT and its variants hold great promise for the future of musculoskeletal imaging. This review will primarily focus on the applications of dual-energy CT in various musculoskeletal pathologies.

## 4. Gout

The prevalence of gout has increased significantly over the years, with over 12 million reported cases in the United States in 2017–2018 alone [7]. Gout is characterized by the deposition of monosodium urate crystals, causing intense pain, erythema, and swelling in the affected area [8]. Gout remains an enormous societal healthcare burden and is associated with a variety of co-morbid conditions, including hypertension, obesity, coronary artery disease, diabetes mellitus, and metabolic syndrome [9]. Currently, the mainstay in diagnosing gout involves visualization of negatively birefringent crystals under polarizing light microscopy. Yet, over the past decade, DECT has become integrated into clinical practice to definitively diagnose and follow up gouty arthritis. In fact, DECT is incorporated in the 2023 European League Against Rheumatism (EULAR) diagnostic criteria for gout [10]. This technique noninvasively visualizes, quantifies, and characterizes monosodium urate crystals by material decomposition, scanning at two different energies simultaneously and differentiating the chemical composition based on atomic number [11]. In patients with established gout, this technique has high diagnostic accuracy, with certain meta-analyses reporting pooled sensitivities and specificities to be 0.81 (0.77, 0.86) and 0.91 (0.85, 0.95), respectively [12]. Figure 1 showcases the axial CT and color-coded reconstruction of monosodium urate crystal deposition in a patient with known gouty arthritis. In patients without a previously established diagnosis, DECT can be beneficial in providing a definitive diagnosis when joint aspiration is negative for monosodium urate crystals. Bongartz et al. reported that DECT correctly identified gout in almost 50% of patients lacking crystals on a synovial fluid analysis and without a prior gout diagnosis [13].

Image postprocessing methods such as virtual monochromatic images (VMI) and three-material decomposition can also improve the diagnostic capability of DECT in gouty arthritis [14]. The mechanism behind VMI involves reconstructing DECT images with different photon energy levels and extrapolating them as if they were obtained at a single energy level, thus decreasing metal artifacts [15]. Three-material decomposition is a postprocessing technique that decomposes images into sets of three materials, allowing for their differentiation and quantification [16]. In a phantom study by Tse et al., the authors reported that the combined use of three-material decomposition and VMI allowed for the detection of monosodium urate crystals at lower concentrations [14].

Another advantage of DECT is its ability to quantitatively evaluate tophaceous deposits, thus making it highly useful in assessing treatment response [17]. One study evaluated the tophi burden on DECT before and after pegloticase, a uric acid-lowering therapy. The authors found that pegloticase decreased the tophi burden by over 70% over 13 weeks [18]. Interestingly, articular tophi resolved quickly over this time frame, whereas tendon tophi took longer to respond. Figure 2 demonstrates DECT images of a patient with tophaceous gout before and after uric acid-lowering therapy. In joints like the spine or hip, which are more difficult to evaluate via physical examination or other non-imaging modalities, DECT may be particularly useful in monitoring treatment response [19]. Yet, DECT is inherently prone to “clumpy artifacts” which can confound diagnostic efforts and incorrectly detect gouty tophi [20]. One study incorrectly noted the presence of gouty crystals in 23–55% of patients without gout, though the incidence of these artifacts can be reduced by increasing the software attenuation value and with the use of a tin filter [21]. Other challenges associated with the use of DECT in evaluating the tophi burden and treatment response are evident in patients with recent-onset gout, a low urate level, or non-tophaceous gout [22]. In a meta-analysis by Gamala et al., the pooled (95% confidence interval (CI)) sensitivity and specificity at the joint level were 0.55 (0.46–0.64) and 0.89 (0.84–0.94), respectively, at a disease duration less than six weeks [10]. Jia et al. further confirmed these findings by reporting the sensitivity of DECT to be 0.36 in patients undergoing DECT for an initial flare [23]. In contrast, the sensitivity for disease at more than 24 months was 0.93. In a small series of 11 patients with non-tophaceous gout, DECT only detected deposits in 3 of 12 joints noted to have monosodium urate crystals at joint aspiration [24]. Using iodinated contrast media can partially overcome these limitations and improve the detection of small uric acid deposits, though higher contrast concentrations can obscure tophi and complicate monosodium urate crystal detection [25]. Still, further research is necessary to improve the sensitivity of DECT in patients with early-onset disease and non-tophaceous gout.

## 5. Inflammatory Arthropathies

Dual-energy CT can also mitigate the diagnostic dilemma between gout and inflammatory arthropathies such as rheumatoid arthritis (RA). When chronic tophaceous gout goes untreated, it can yield chronic polyarthritis, which can be an RA mimicker [26]. In a case presentation by Sanghavi et al., the authors discuss a patient presenting with chronic bilateral joint pain and X-ray findings of periarticular erosions, small joint effusions with synovitis, and marked soft-tissue edema interpreted as an inflammatory arthropathy [27]. The diagnosis of an inflammatory arthropathy was not clear cut, as the patient denied significant morning stiffness, and both the rheumatoid factor and anti-cyclic citrullinated peptide (CCP) antibodies were negative. Due to this diagnostic uncertainty, the patient underwent DECT of the right wrist, as it was unclear whether this was seronegative RA, and the effusions were not large enough to obtain an arthrocentesis. The DECT of the right wrist ultimately showed uric acid deposition within the triangular fibrocartilage complex and scapholunate ligament. Other authors have also reported the utility of DECT in evaluating rheumatoid arthritis. Jans et al. noted that DECT has certain advantages over traditional MR imaging. These authors state that DECT directly portrays the cortex and thus more reliably detects bony destruction and erosions at a lower cost and with a higher accessibility than conventional MRI, with high agreement between both modalities [28].

In addition to distinguishing gout from rheumatoid arthritis, DECT is useful in differentiating gout from pseudogout, also known as calcium pyrophosphate deposition disease (CPPD). DECT is particularly beneficial in assessing atypical presentations of CPPD or sites where aspiration is unavailable, as gout and pseudogout can be clinically indistinguishable in the hands and wrist [29]. Figure 3 demonstrates a scenario in which DECT differentiated between monosodium urate crystals and calcium pyrophosphate crystals in the hand [30]. Collagen density mapping can also provide information regarding the ligament integrity of patients with CPPD. The mechanism behind collagen mapping lies in the varying attenuation of the hydroxyproline and hydroxylysine molecules within a collagen side chain, providing vital information about soft-tissue structures [29]. Interestingly, Ziegeler et al. reported that the collagen density of the scapholunate ligament was significantly higher in patients with CPPD compared to controls [31]. Although it has been postulated that crystal deposition would decrease the extracellular matrix, thus destabilizing the ligament, this study emphasized the fact that the extracellular matrix remodels in response to crystal deposition [31]. Ultimately, DECT has high clinical utility in providing the correct diagnosis and yielding vital information on the molecular level.

Psoriatic arthritis (PsA) is a chronic immune-mediated inflammatory arthropathy that affects 125 million people globally [32]. With the introduction of novel biologic medications, imaging techniques such as contrast-enhanced DECT can play an essential role in monitoring synovitis and enthesitis throughout treatment (Figure 4). While MRI is often utilized to evaluate psoriatic joint involvement, this technique has several disadvantages. First, MRI tends to produce significant artifacts in the small joints of the hand, such as the distal interphalangeal (DIP) joints, joints often affected in PsA. Furthermore, MRI has inadequate spatial resolution when evaluating small joints [33]. In a study by Fukuda et al., the authors noted that DECT detected extensor peri-tendonitis, synovitis, and periarticular inflammation significantly more often than on an MRI [34].

DECT with iodine mapping has also uncovered findings not discovered by other imaging techniques in the realm of psoriatic arthritis. In a case of PsA associated with nail psoriasis, DECT with iodine mapping was able to detect active enthesitis in the nail root where X-ray and dermatoscopy of the hyponychium could not [35]. In another study by Umezawa et al., the use of iodine mapping with DECT uncovered a case of psoriatic arthritis in the knee by detecting active enthesitis not evident on prior conventional CT and X-ray images [36]. Ultimately, the image-processing technique of iodine mapping can improve iodine contrast resolution and more clearly illustrate the various patterns of inflammation in psoriatic arthritis on DECT [37].

Other studies have used DECT iodine maps to semi-quantitate inflammation in patients with symptomatic and non-symptomatic PsA of the distal joints before and after biologic treatment [33]. These authors ultimately found decreased iodine uptake after biologic treatment and an overall improvement in the PsA DECT iodine map scoring system (PsADECTS), which included DECT evaluation of synovitis, extensor peri-tendonitis, periarticular inflammation, and flexor tenosynovitis [33]. While DECT may have significant value in the quantitative assessment of the therapeutic response to psoriatic arthritis, it is essential to note that this was a post-contrast study, as the utility of DECT without contrast is limited when evaluating synovitis and tenosynovitis.

## 6. Bone-Marrow Edema and Fracture Identification

Recent studies have also shown the clinical utility of DECT in assessing bone-marrow edema to identify and characterize fractures (Table 2). Bone-marrow edema is a non-specific finding associated with fractures, inflammation, and other acute injuries. While MRI is the first-line imaging modality in assessing bone-marrow edema, some patients have contraindications to this technique, such as non-compatible pacemakers, neurostimulators, or other implants [38]. Further, this technique is costly, timely, and not readily available in low-resource settings. In addition, the rapid acquisition time and absence of motion artifacts associated with DECT make it a favorable option in emergent settings [38].

In order to evaluate bone-marrow edema most effectively, virtual non-calcium reconstruction algorithms (VNCa) can be utilized. This post-processing technique suppresses the high attenuation of trabecular bone by reducing or removing the presence of calcium, thus allowing for better visualization of the underlying marrow [38]. In many patients with underlying skeletal disorders, such as osteoporosis or Paget’s disease of bone, subtle femur or pelvic fractures may go undetected on conventional radiographs or standard CT imaging [39]. In a study by Jang et al., the authors noted that the sensitivities and specificities of VNCa images in detecting non-displaced proximal femur and pelvic fractures were 100% and 94%, respectively [39]. Based on the high sensitivity of this technique, radiologists and patients can be easily reassured that osseous injuries are excluded when bone-marrow edema is absent, eliminating the need for further imaging studies.

For certain anatomic regions such as the wrist, fractures often go undetected, leading to complications such as osteonecrosis, osteoarthritis, or nonunion. Certain studies indicate that the sensitivity of conventional radiographs in detecting small fractures only reaches 59–79%, making this technique unreliable [49]. Further, Hunter et al. note that only 16% of scaphoid fractures are visible on initial radiographs [50]. In a retrospective study assessing wrist fractures, the authors noted that DECT could identify bone-marrow edema and thus detect wrist fractures with a sensitivity and specificity of 100% and 99.5%, respectively [40]. Figure 5 showcases a non-displaced scaphoid fracture identified on DECT that was not suggestive of fracture on a standard radiograph.

DECT can also be utilized in evaluating vertebral fractures as well. Recent studies have noted that DECT has sensitivities and specificities of 84% and 98% when assessing the spine and appendicular skeleton [41]. A separate pooled analysis of DECT diagnosis of vertebral compression fractures calculated similar sensitivities and specificities of 82% and 98% respectively [43]. Certain studies report that the high specificity of this technique may reduce the need for confirmatory MRI in emergent settings [42]. Karaca et al. also noted the utility of DECT in distinguishing between acute and chronic vertebral fractures in patients incompatible with MR imaging [44]. Certain authors also note that MRI provides significantly lower diagnostic confidence in detecting acute fracture lines when compared with DECT, which may result in false-negative readings and an underestimation of vertebral fractures [46]. Figure 6 compares DECT and MRI in the assessment of fracture lines.

## 7. Acute Knee Trauma

In addition to occult fracture detection, DECT virtual non-calcium techniques can provide information regarding soft-tissue integrity, as some authors report that the shape and location of bone-marrow edema present can predict ligamentous and meniscal injury after acute knee trauma [47]. The diagnostic capability of DECT in detecting acute knee injury has also been evident across meta-analyses, with pooled sensitivities of 84% and 85%, and specificities of 96% [48,52]. In anterior cruciate ligament (ACL) tears, DECT can demonstrate focal areas of bone-marrow edema corresponding to bone contusions [38]. As most ACL tears are associated with tibial plateau trabecular contusions and lateral femoral condyle contusions, identifying bone-marrow edema in these regions can correlate to ligamentous injury [38]. While these traditional contusion patterns can help clinicians make the correct diagnosis, collagen mapping can also provide insight into soft tissue integrity. Gruenewald et al. noted that color-coded collagen reconstructions derived from DECT images yielded higher diagnostic accuracy and image quality in assessing ligamentous injury compared to a standard grayscale CT [53]. Other authors reported similar findings, noting that DECT had sensitivities and specificities of 97.1% and 98%, respectively, in detecting ACL rupture [54]. Although both studies note that DECT concurred with MRI findings, DECT images are often of lower quality compared to MRI due to lower contrast-to-noise and signal-to-noise ratios [29,55].

## 8. Degenerative Disc Disease

Degenerative disc disease (DDD) is a significant cause of disability worldwide, and its prevalence has increased considerably over the years. Histopathologically, one of the hallmarks of degenerative disc disease is the “dehydration” of the nucleus pulposus due to loss of proteoglycan content [56]. While certain authors have utilized VNCa to characterize disc herniation, few studies have examined its application in degenerative disc disease. It has been hypothesized that color-coded VNCa reconstructions can reflect the water content within intervertebral discs and thus grade the severity of degenerative disc disease [56]. This is accomplished via the ability of DECT to detect the increase in disc attenuation that correlates to the decrease in proteoglycan content, resulting in nucleus pulposus dehydration and disc height reduction [51]. Figure 7 demonstrates a color-coded VNCa reconstruction of the spine.

## 9. Osteoporosis

DECT has also been utilized in the opportunistic evaluation of bone mineral density (BMD) and fracture risk evaluation. Osteoporosis, also known as a decrease in BMD, is a highly prevalent metabolic bone disease, affecting one in three women and one in five men globally [57]. At present, dual-energy X-ray absorptiometry (DXA) scanning remains the gold standard for diagnosis and treatment monitoring of osteoporosis [58]. Yet, many studies note that DXA scanning is an underutilized technique, with screening rates as low as 30% in Medicare patients [59]. Further, there are many limitations to traditional DXA scanning, as overlying tissue, bowel contents, and vascular calcifications can yield variations in BMD values [60]. Although quantitative CT (QCT) has also been used to evaluate BMD, this technique is costly, affected by marrow adipose tissue, and provides higher radiation-dose exposure than DXA scanning [61]. Thus, DECT may have a role in evaluating BMD and fracture risk in patients already undergoing scans, as material decomposition allows for BMD measurements with minimal influence of marrow adipose tissue. Further, hydroxyapatite (HAP) concentrations can effectively evaluate trabecular bone and allow for volumetric BMD quantification [62]. Yet, there is currently mixed data on the accuracy of DECT in the assessment of BMD. Certain studies state that DECT-associated HAP values moderately correlate with DXA images (r = 0.614) while others report linear correlations (r = 0.97) [60,63]. Yet, it is imperative to note that the latter study used a phantom setup and has yet to validate their methods in a clinical setting. Other authors report the utility of DECT in assessing fracture risk and state that DECT-based volumetric BMD values can predict 2-year fracture risk with sensitivities and specificities of 85.45% and 89.19%, respectively [64]. Ultimately, further research is necessary before DECT can be considered a standard of care for the diagnosis of osteoporosis. However, clinicians should consider leveraging DECT to evaluate bone mineral density opportunistically in patients undergoing imaging procedures.

## 10. Primary Malignant Bone Cancers

Other applications of DECT include identifying bone-marrow infiltration of primary malignant bone cancers (PMBCs). PMBCs are often aggressive tumors that can invade surrounding bone and the adjacent bone marrow; thus, it is imperative that they are evaluated and staged early. As bone-marrow composition changes with tumoral invasion, DECT can sensitively detect abnormal bone-marrow density [65].

Interestingly, recent studies have looked at the utility of biosynthetic nanosheets (BiOI NS) as a DECT contrast agent to enhance this technique’s clarity and accuracy in evaluating osteosarcoma [66]. Certain authors note that using BiOI NS in DECT yielded higher diagnostic accuracy in differentiating osteosarcoma from healthy bone [66]. Other studies have evaluated DECT-guided photothermal therapy, using BiOI NS as photothermal agents in order to target cancer cells more precisely. In a study by Li et al., the authors reported that DECT can showcase the real-time diffusion of various nanomaterials into a tumor and provide information on the distribution of the treatment to avoid normal tissue injury [45]. While osteosarcoma has long been considered non-sensitive to radiotherapy, the advent of DECT and BiOI NS can allow a larger X-ray dose to be deposited into the tissue and may be able to increase the tumor’s sensitivity to this technique [45]. Ultimately, further studies are indicated to evaluate the role of DECT-guided radiotherapy and photothermal therapy, but these techniques hold great promise for the future identification and treatment of osteosarcoma.

Other authors have looked at the use of DECT in detecting bone-marrow involvement in multiple myeloma (MM). Traditional CT imaging has primarily been used to detect osteolytic lesions, with MRI as the gold standard for determining bone-marrow involvement [67]. Yet, the advent of DECT allows more precise identification of the tumor burden, degree of infiltration, and treatment response. Werner et al. retrospectively evaluated spinal osteolytic lesions in 32 patients with known MM to differentiate between active and inactive lesions [68]. Ultimately, the authors found that the level of VNCa attenuation differs between active and inactive diseases and can shed light on tumor activity [68]. DECT also has the potential to detect bone-marrow infiltration with high sensitivity [69]. In a study by Fervers et al., the authors noted that a non-adipose segment of bone marrow predicted bone-marrow infiltration when using artificial intelligence (AI)-supported assessment of DECT VNCa attenuation values [70] (Figure 8). Further, the authors noted that a non-adipose fraction of bone marrow > 0.93% could predict an osteolytic lesion in that region [70]. As bony involvement is a common manifestation of MM and reflects advanced disease, the use of DECT may play an essential role in analyzing the disease extent and identifying the risk of progression.

## 11. Bone Metastases

In addition to imaging primary malignant bone cancers, DECT can also evaluate bone metastases. Not only is bone the third most common site of metastases, but metastases are the most common cause of bone tumors [71]. The imaging of bone metastases is crucial for cancer staging, and consequently, for developing an appropriate treatment plan. In such applications, DECT has demonstrated strong performance in differentiating malignant tumors from non-malignant lesions by utilizing quantitative parameters. Xu et al. derived quantitative features from DECT such as dual-energy index, atomic number, regular CT, and electron density to differentiate between osteoblastic metastases and bony islands, a benign sclerotic osseous lesion that often mimics a metastatic lesion [72]. Dong et al. also tried to differentiate bony islands and osteoblastic metastases on DECT and noted that the mean attenuation values were much higher in osteoblastic metastases than in bony islands [73]. DECT can also play a role in directing bone biopsies for metastatic lesions. In such settings, DECT guidance is less expensive and faster than positron emission tomography (PET)/CT guidance and does not require specialized devices nor result in the large artifacts often associated with MRI guidance [74]. Thus, DECT holds great promise in the diagnosis and differentiation of bony metastases.

Using postprocessing techniques such as VNCa can further enhance the capabilities of DECT in imaging bone metastases. Abdullayev et al. reported that VNCa, with low to intermediate calcium suppression indices, was most effective in differentiating normal and metastatic bone [75]. In the detection of osteolytic lesions in non-small cell lung cancer across two readers, DECT with VNCa images had higher specificity than ^18^F-fluorodeoxyglucose (18F-FDG) PET/CT (94.1%/94.1% vs. 82.4%/76.5%), but lower sensitivity (80%/76% vs. 96%/100%) [76]. Thus, DECT complements traditional PET/CT in reducing the prevalence of false-negative findings. In another report, using color-coded bone-marrow maps enhanced the sensitivity, specificity, accuracy, and confidence of readers in detecting bone metastases [77]. However, the authors noted that these maps are less effective in patients with high red-marrow content, such as younger patients [77]. Ultimately, the role of DECT in the detection and treatment of bone metastases is still emerging.

## 12. Metal-Artifact Reduction

In orthopedic imaging, VMI and metal-artifact reduction (MAR) algorithms can significantly improve the visualization of anatomy and soft tissue after total hip arthroplasty (THA) or other implants, including those of the radius, spine, and shoulder (Table 3). Traditional CT techniques are often limited by three different types of artifacts: scatter artifacts, photon starvation, and beam hardening [78]. The first, scatter artifacts, occur because of the large difference in attenuation between soft tissue and an adjacent metal implant [79]. Artifacts such as photon starvation and beam hardening can also arise due to the absorption of all photons (caused by the high attenuation of metal) and absorption of low energy photons, respectively, distorting the image [80]. Interestingly, certain implant types generate more artifacts than others. For instance, Wellenberg et al. reported that stainless-steel implants created far more metal artifacts than titanium implants due to the higher molecular weight of stainless steel [81]. Thus, the optimal monochromatic energy of a titanium tibia plate and a stainless-steel tibia plate were 130 keV and 180 keV, respectively [81]. Thus, clinicians must consider the type of implant used in order to maximize MAR, diagnostic accuracy, and overall image quality.

The size of the implant and the keV level also influence the amount of metal artifacts produced. In a study by Kosmas et al., the authors evaluated the utility of DECT in reducing artifacts generated by both small and large metal implants compared to traditional CT techniques [78]. These researchers found that virtual monochromatic reconstructions had higher diagnostic quality and a reduced level of metal artifacts in smaller implant sizes [78]. They also reported a progressive decrease in artifact size at higher keV levels (Figure 9). Yet, it has been noted that there is decreased vascular visualization at higher keV levels [82]. Zhao et al. stated that vascular visualization of periprosthetic vessels in THA patients decreased with the increasing energy level despite improved metal-artifact reduction and noted that 80 keV provides the optimal balance between MAR and vascular visualization [83]. Thus, in larger implants, high keV VMI may only yield limited artifact reduction in addition to decreased vascular attenuation [90].

To mitigate such limitations, iterative metal-artifact reduction (iMAR) algorithms can be utilized at lower energy levels. This allows for vascular visualization and minimizes the large number of metal artifacts typically associated with lower energy levels [84]. iMAR algorithms use an iterative loop to identify areas in the image affected by metal hardware, estimate what the image should look like without the metal hardware, and then correct and refine the image to reduce metal artifacts [91]. The combination of these iMAR algorithms with VMI has also shown great promise in musculoskeletal imaging. A study by Laukamp et al. reported that combining iMAR and VMI yielded significant metal-artifact reduction, as these techniques complement one another. VMI is beneficial in evaluating periprosthetic bone, while iMAR is the preferred method for soft-tissue analysis [79]. Additionally, though not classified as DECT, recent studies have also utilized photon-counting CT for more precise energy resolution and detection [92].

## 13. Limitations of DECT

While DECT has many strengths within musculoskeletal imaging, this technique also has some limitations. As mentioned in the gout section, DECT is non-sensitive in patients with a low urate level, recent-onset gout, or non-tophaceous gout [22]. Further, incorrect attenuation thresholds in material-decomposition algorithms can yield false-negative or positive results when depicting tophi [93]. While DECT can reduce metal artifacts, it is also prone to artifacts itself, some of which are due to post-processing techniques, and others are inherent to the scanner’s design. Although DECT is very effective in differentiating materials at different energy levels, this technique has difficulty distinguishing materials with similar attenuation, leading to possible misinterpretation [93]. Moreover, certain authors note differences between the material-decomposition methods among DECT manufacturers, leading to high variability and low reproducibility when using different DECT scanners [93,94]. Lastly, DECT is non-validated in various musculoskeletal conditions compared to more established modalities like MRI. Thus, future studies are necessary to establish the accuracy and reliability of DECT in clinical practice.

## 14. Conclusions

Ultimately, DECT has excellent utility in diagnosing and treating various musculoskeletal conditions, particularly in patients with contraindications to MRI. This technique’s ability to identify subtle compositional changes and differentiate between multiple tissue types holds significant promise to delineate bone-marrow edema, malignant invasion, radiographically occult or subtle fractures and contusions, improve metal-artifact reduction, and mitigate diagnostic dilemmas between inflammatory arthropathies and gout. In addition, post-processing techniques, such as VNCa reconstructions, VMI, iMAR algorithms, and collagen density mapping, allow better visualization of the underlying bone-marrow and soft-tissue integrity than traditional imaging modalities. As technology continues to evolve, DECT may change the future of clinical decision-making, particularly with advancements in guiding radiotherapy and photothermal therapy in primary malignant bone cancers and predicting osteolytic lesions in multiple myeloma. DECT’s integration into clinical practice and ongoing research and advancements solidifies DECT as an indispensable modality, enabling physicians to navigate complex musculoskeletal pathologies with unparalleled accuracy and efficacy.

## Figures and Tables

**Figure 1 diagnostics-14-00732-f001:**
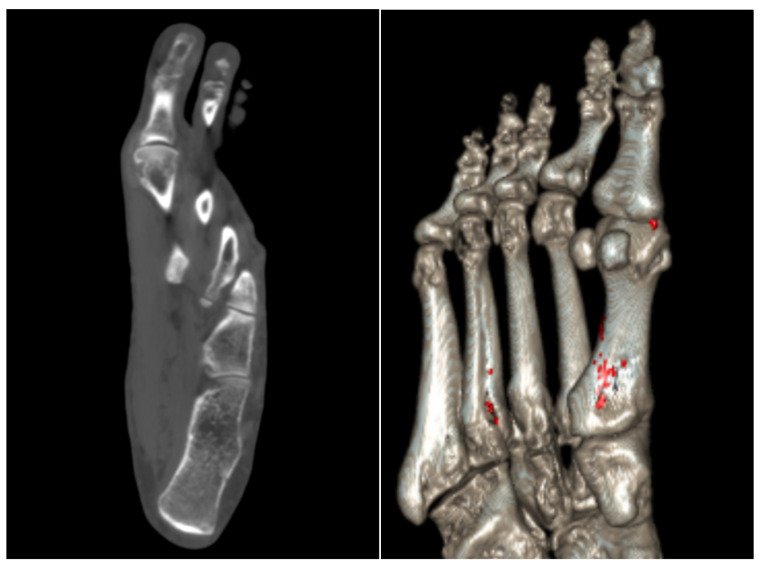
Axial CT of the foot (**left**) in a patient with known gouty arthritis demonstrating an erosion at the first metatarsal head with adjacent faint soft-tissue tophus. Color-coded reconstruction (**right**) shows monosodium urate crystal deposition medial to the 1st metatarsophalangeal joint, corresponding to the soft tissue tophus on the left image. Increased color involving the cortex of the 1st and 4th proximal metatarsals felt to be artifactual.

**Figure 2 diagnostics-14-00732-f002:**
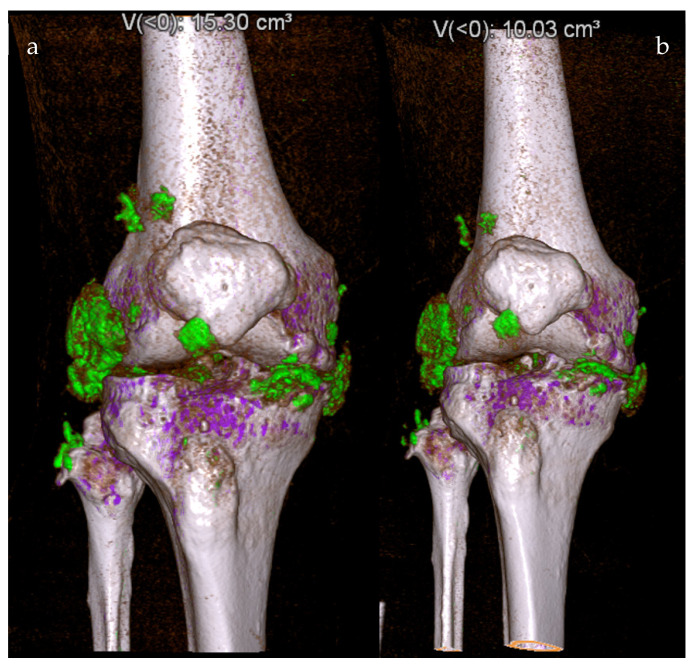
DECT color-coded 3D reconstructions showing a patient with tophaceous gout of the right knee before (**a**) and eight weeks after treatment with pegloticase (**b**), with a 34% reduction in monosodium urate (MSU) deposition, from 15.3 cm^3^ to 10.03 cm^3^. Purple: calcium; Green: gout deposits.

**Figure 3 diagnostics-14-00732-f003:**
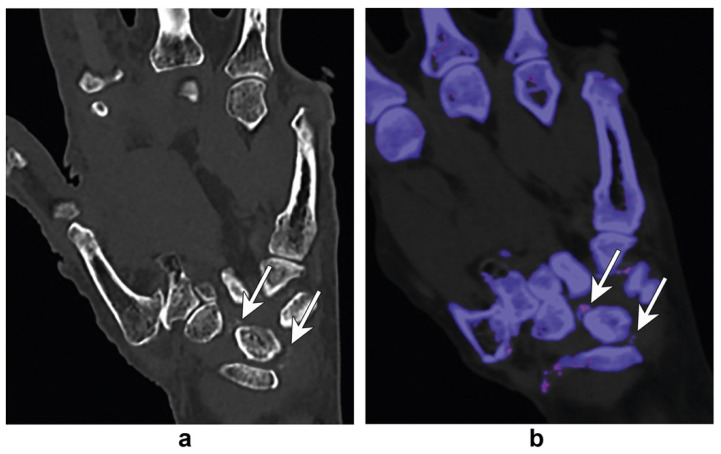
Standard coronal CT of the hand (**a**) demonstrates dense foci in the intercarpal region (arrows in (**a**)). Differential diagnoses include gout or CPPD. DECT (**b**) demonstrates the absence of gout crystals, and the dense foci are color-coded (blue) similar to bone, which confirms CPPD. Figure was reproduced with permission from Gandikota et al., [30].

**Figure 4 diagnostics-14-00732-f004:**
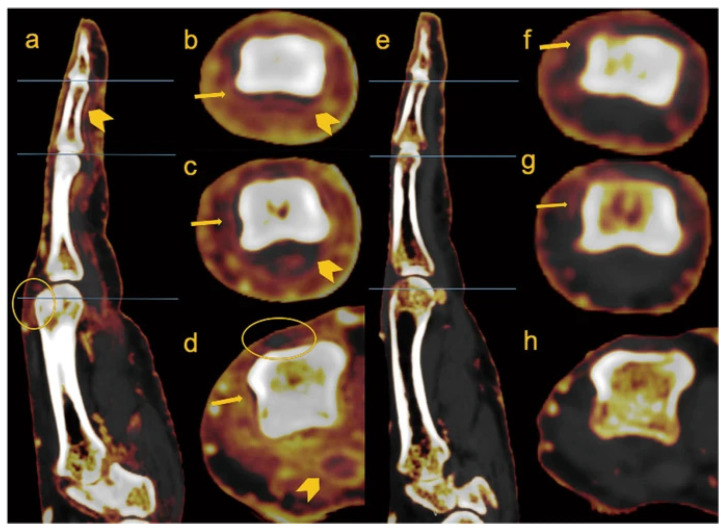
Sagittal and axial images of the right 2nd digit, pre (**a**–**d**) and 1-year post-treatment (**e**–**h**) with Adalimumab, of 80-year-old male PsA patient. Circular enhancement along the joint capsule suggests synovitis (arrows in (**b**–**d**)), flexor tenosynovitis (arrowheads in (**a**–**d**)), and obvious periarticular enhancement is seen before treatment. Extensor peritendonitis (circle in (**a**)) is also seen on the MCP, pre-treatment. Those findings improved on the post-treatment with only mild synovitis remaining (arrows in (**f**,**g**)). Figure reproduced with permission from Kayama et al. [33].

**Figure 5 diagnostics-14-00732-f005:**
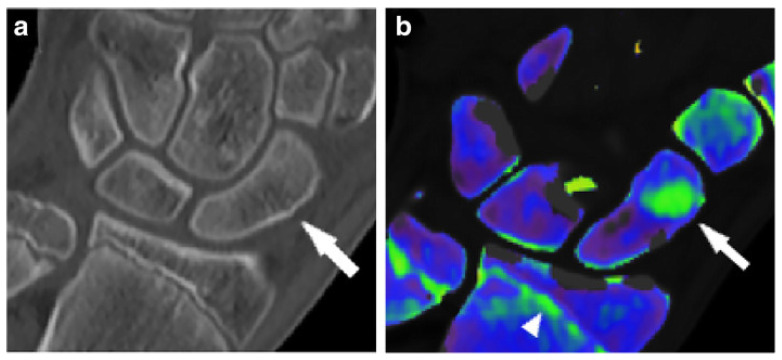
Hand dual-energy computed tomography scan in a patient with a non-displaced scaphoid fracture, confirmed with magnetic resonance imaging. Standard grayscale series (**a**) shows a subtle cortical interruption, which is not suggestive of fracture (arrow). A color-coded virtual non-calcium image (**b**) depicts the presence of bone-marrow edema, confirming the hypothesis of a traumatic lesion. Notably, the epiphyseal line on the distal radius and ulna are also color-coded in green (arrowhead). Figure was reproduced with permission from D’Angelo et al. (2021) [51].

**Figure 6 diagnostics-14-00732-f006:**
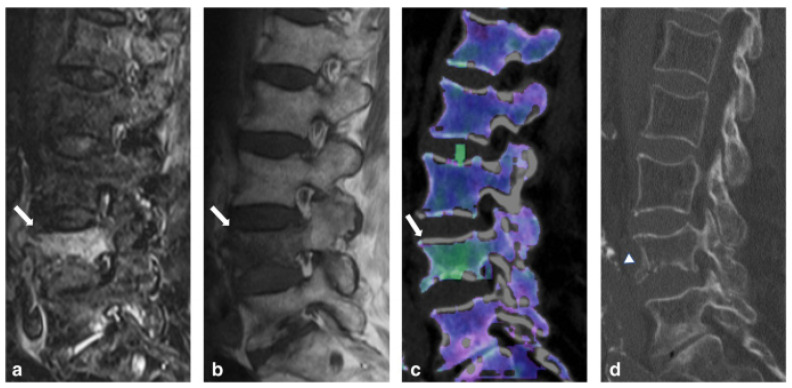
An 87-year-old woman presenting with acute spinal trauma due to a domestic fall. (**a**) Sagittal turbo inversion recovery magnitude (TIRM)–magnetic resonance imaging (MRI) series, (**b**) spin-echo (SE) T1-weighted MRI series, and (**c**) dual-energy computed tomography (DECT)–virtual non-calcium (VNCa) reconstructions showing bone-marrow edema (BME) in all four quadrants of L4 (arrow). All readers were concordant in assessing BME presence (score 3 = distinct BME) and extent (score 4 = all quadrants) with both techniques. (**d**) In addition, sagittal conventional grayscale DECT images allowed for the detection of an acute slightly dislocated fracture of the ventral ground plate of L1 (arrowhead) in terms of a teardrop fracture with potentially associated instability by all readers in this study. Confidence in depicting fracture lines was rated as intermediate (score 2) and high (score 3) by 5/5 readers on the MR and DECT image series, respectively. Figure was reproduced with permission from Cavallaro et al. [46].

**Figure 7 diagnostics-14-00732-f007:**
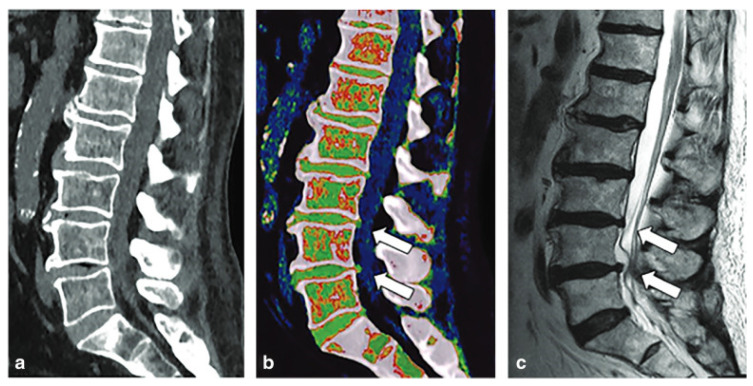
Spine dual-energy computed tomography. Standard grayscale series (**a**) shows typical findings of spondylarthrosis with vacuum phenomena in L3/L4 and L4/L5 intervertebral discs. Virtual non-calcium reconstruction with optimization for intervertebral disc analysis (**b**) can finely show the protrusion of lumbar discs (arrows), confirmed by magnetic resonance imaging T2-weighted sequence (**c**) (arrows). Figure was reproduced with permission from D’Angelo et al. (2021) [51].

**Figure 8 diagnostics-14-00732-f008:**
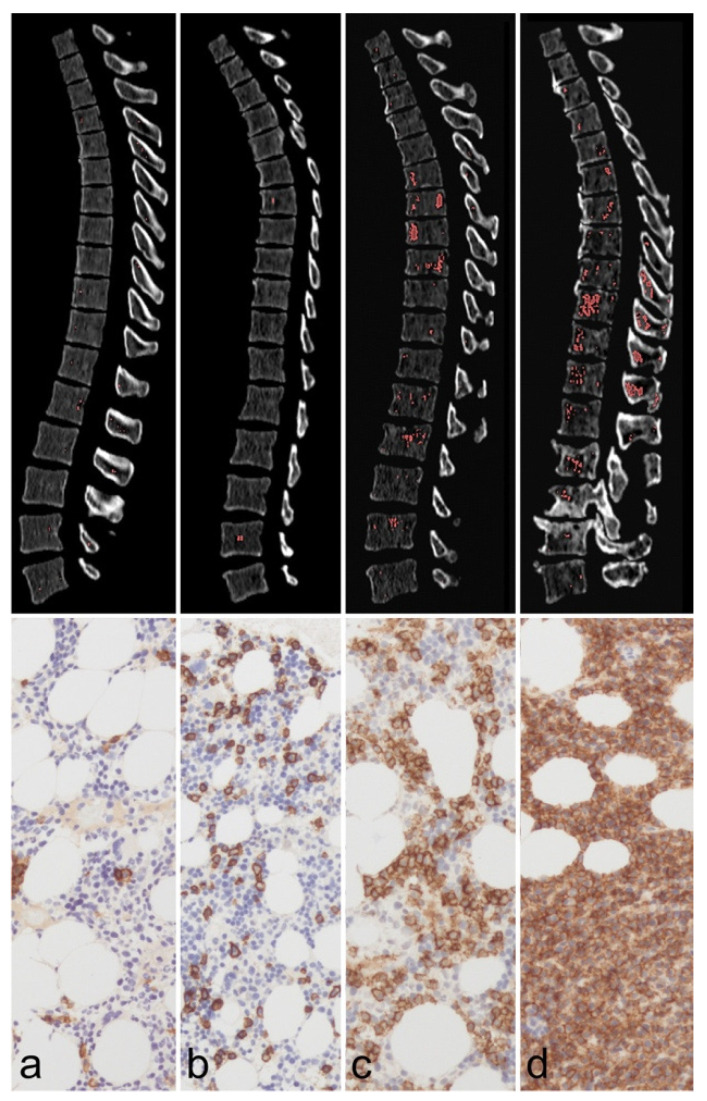
Spine infiltration by multiple myeloma (MM) displacing the fatty bone marrow (BM). Four patients (**a**–**d**) with parallel dual-energy CT (DECT, top row) and CD138-immunostained BM biopsy in 200 × amplification (bottom row) are presented as an example. The thoracolumbar spine was automatically segmented by a convolutional neural network. The red overlay on the DECT slices marks voxels with an attenuation > 0 HU in post-processed virtual non-calcium (VNCa) images. The percentage of BM infiltration, as determined by biopsy, was 0–5%, 10–15%, 60–70%, and 90% for patients (**a**–**d**), respectively. With rising BM infiltration, an increase of the non-fatty attenuating portion of BM on VNCa images is visually assessable (larger patches of red overlay) and measurable (0.3%, 0.4%, 4.3%, and 5.4% for patients (**a**–**d**), respectively). Correspondingly, the histological images demonstrate an expansion of CD138 +, brownish-stained plasma cells, and a displacement of the translucent, fatty vacuoles. We hypothesize that these histological findings correspond to the rise of attenuation, which we observed on VNCa BM data. Figure reproduced with permission from Fervers et al. [70].

**Figure 9 diagnostics-14-00732-f009:**
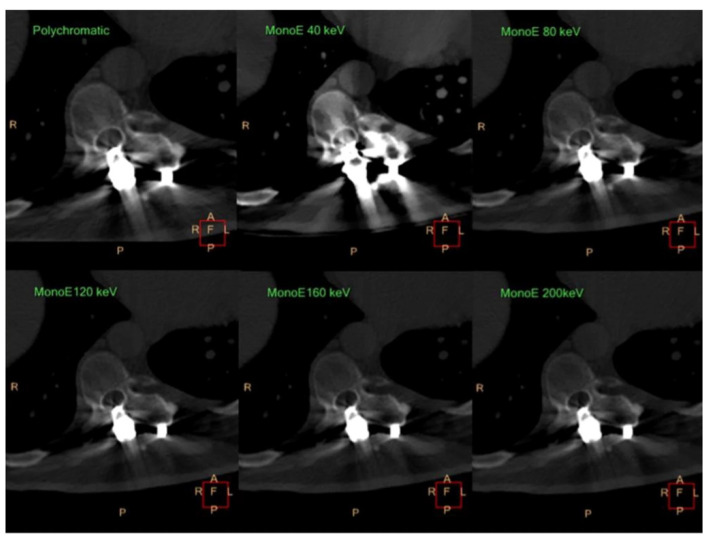
Progressive reduction of artifact size in virtual MonoE spectral CT images compared to the conventional polychromatic images (60, 100, 140, and 180 keV images are omitted). Figure reproduced with permission from Kosmas et al. [78].

**Table 1 diagnostics-14-00732-t001:** **Spectral CT Imaging Modalities.** An overview of spectral CT imaging modalities and subtypes, including notable strengths and weaknesses.

Modality	Image Acquisition Approach	Strengths	Weaknesses
**Dual Energy:**			
**(a) Dual Spin**	Source-based; Utilizes a single X-ray source with two scans conducted sequentially at different energy levels.	Least complex hardware requirements	Sequential nature of scans leads to poor temporal resolution, making this technique especially weak in imaging of movable anatomical structures (e.g., the heart or respiratory structures); Higher radiation dose.
**(b) Dual Source**	Source based Employs two separate X-ray sources and detectors at differing energy levels	High- and low-energy images can be simultaneously acquired, providing extremely high temporal resolution; High spectral separation (high contrast)	The dual-energy mode must be proactively; chosen before imaging. The detectors must be of differing sizes, with one smaller than the other, leading to field-of-view limitations.
**Rapid Kilovolt Peak Switching**	Source based; Utilizes just one detector with the X-ray source alternating rapidly (more than every millisecond) between high and low energy levels during scanning.	Near-simultaneous acquisition of high- and low-energy images	Alternating of energy levels leads to low spectral separation
**Split-Filter Twin Beam**	Source based; Utilizes a single X-ray source with a two-material filter split beam to produce differing energy spectra.	Many existing CT scanners can be retrofitted with a split filter.	Limited spectral separation
**Dual-Layer Computed Tomography**	Detection-based; Utilizes one X-ray source with two layered or “sandwiched” detectors that collect high- and low-energy data, respectively.	Images from both energy levels are acquired simultaneously and additionally. Spectral imaging information can be reconstructed after imaging.	Low spectral separation
**Photon-Counting Computed Tomography**	Directly counts individual photons at varying energy levels by converting X-ray energy to an electronic signal	Images offer superior contrast and resolution. Lower radiation doses Spectral imaging information can be reconstructed after imaging.	Limited availability

**Table 2 diagnostics-14-00732-t002:** Studies assessing diagnostic capabilities of DECT in bone-marrow edema and fracture identification.

Reference	Condition Being Diagnosed	Dual-Energy CT Type	Sample Size (Patients)	Sensitivity	Specificity
Jang et al., 2019 [39]	Nondisplaced fractures of the hip (proximal femur and pelvic bones)	Dual-Source CT System (SOMATOM Definition Flash, Siemens Healthcare)	35	100%	94%
Ali et al., 2018 [40]	Acute fractures of the wrist (carpal bones)	Dual-Source CT System (SOMATOM Definition Flash, Siemens Healthcare)	37 (24 visually interpreted to develop an attenuation value cutoff; 13 to validate the cutoff)	100% (develop-mental set); 100% (validation set)	99.5% (develop-mental set); 100% (validation set)
Suh et al., 2018 [41]	Bone-marrow edema of the spine, ankle, hip, and knee	Various (Meta-analysis)	450 (Meta-analysis)	85% (overall performance)	97% (overall performance)
Sherbaf et al., 2021 [42]	Vertebral fractures	Various (Meta-analysis)	606 (Meta-analysis)	89% (overall performance)	96% (overall performance)
Yang et al., 2018 [43]	Vertebral compression fractures	Various (Meta-analysis)	510 vertebrae (Meta-analysis)	82%	98%
Karaca et al., 2016 [44]	Vertebral compression fractures	Dual-Source CT System (SOMATOM Definition Flash, Siemens Healthcare)	23	89.3%	98.7%
Li et al., 2023 [45]	Acute knee injury	Various (Meta-analysis)	290 (Meta-analysis)	85% (overall performance)	96% (overall performance)
Cavallaro et al., 2022 [46]	Acute vertebral fractures	Dual-Source CT System (SOMATOM Force, Siemens Healthcare)	88	89% (injury presence); 84% (injury extent)	98% (injury presence); 98% (injury extent)
Booz et al., 2020 [47]	Acute knee injury	Dual-Source CT System (SOMATOM Force, Siemens Healthcare)	57	94% (qualitative analysis); 95% (quantitative analysis with cutoff of −42 Hounsfield units); 96% (quantitative analysis with cutoff of −51 Hounsfield units)	95% (qualitative analysis); 95% (quantitative analysis with cutoff of −42 Hounsfield units); 97% (quantitative analysis with cutoff of −51 Hounsfield units)
Wilson et al., 2021 [48]	Acute knee injury	Various (Meta-analysis)	267 (Meta-analysis)	84% (overall performance)	96% (overall performance)

**Table 3 diagnostics-14-00732-t003:** Overview of studies of metal-artifact reduction in DECT.

Reference	Study Aims	Dual-Energy CT Type	Key Findings Surrounding Metal-Artifact Reduction
Kosmas et al., 2019 [78]	Comparison of the utility of dual-layer CT vs. conventional polychromatic CT in reduction of artifacts from metal implants	Dual-Layer CT System (Prototype, Phillips Healthcare)	Virtual monochromatic reconstructions offered higher diagnostic quality; Decreased artifact size as energy level increased, with 160 and 180 keV being optimal; Artifact reduction was greater with smaller implants.
Wellenberg et al., 2018 [81]	Quantification and optimization of metal-artifact reduction with virtual monochromatic dual-energy CT for various metal implants, with non-metal scans as a reference for comparison	Dual-Source CT System (SOMATOM Force, Siemens Healthcare)	Stainless-steel implants created more metal artifacts than titanium implants due to higher molecular weight; Optimal monochromatic energies are dependent on the type of metal implant.
Laukamp et al., 2018 [79]	Comparison of metal-artifact reduction of three imaging methods (VMI; metal-artifact-reduction-specialized reconstructions, an iterative algorithm; and conventional images from dual-layer CT) in imaging of total hip replacements	Dual-Layer CT System (IQon, Phillips Healthcare)	VMI and metal-artifact-reduction-specialized reconstructions both resulted in significant artifact reduction compared to conventional images; Relative artifact-reduction performance between VMI and metal-artifact-reduction-specialized-reconstructions differs based on the tissue or anatomical region imaged; Recommended photon energies for VMI range from 140–200 keV.
Neuhaus et al., 2017 [82]	Examination of metal-artifact reduction performance in virtual monoenergetic images from dual-layer CT	Dual-Layer CT System (IQon, Phillips Healthcare)	VMI can significantly reduce artifacts from metal implants; 140 keV generally offered optimal artifact reduction, though, for 20% of patients, individualized photon energies provided improved image quality.
Zhao et al., 2023 [83]	Examination of the image quality of VMI as well as optimization of visualization of orthopedic MAR and the periprosthetic vasculature in THA patients	Dual-Layer CT System (IQon, Phillips Healthcare)	VMI in combination with MAR can reduce metal artifacts and improve vascular visualization; Metal-artifact reduction improved with increasing energy level; Vascular visualization decreased with increasing energy with optimal performance at 80 keV and poor performance above 100 keV.
Wichtmann et al., 2023 [84]	Evaluation of split-filter abdominal CT image quality and metal-artifact reduction using VMI or iMAR in patients with hip or spinal implants	Split-Filter CT System (SOMATOM Definition Edge, Siemens, and SOMATOM Definition AS+, Siemens Healthcare)	VMI at high energy (190 keV) with iMAR displayed the lowest artifact; Overall image quality was highest in non-VMI, mixed images with iMAR; Higher artifacts associated with lower keV could be counteracted by the iMAR algorithm, enabling the use of lower energy images to enhance vascular contrast while limiting artifacts.
Mohammedinejad et al., 2021 [85]	Evaluation of iMAR and VMI in DECT of patients with total shoulder prosthesis	Dual-Source CT System (SOMATOM Force, Siemens Healthcare)	VMI and iMAR independently improved image quality and reduced artifacts, but the strongest performance was achieved by using both
Bongers et al. 2015 [86]	Evaluation of VMI and iMAR in metal-artifact reduction for imaging of hip prosthesis and dental implants	Dual-Source CT System (SOMATOM Definition Flash, Siemens Healthcare)	VMI and iMAR independently reduced artifacts, but the strongest performance was achieved by using both
Long et al., 2019 [87]	Evaluation of VMI, iMAR, and the combination of VMI and iMAR in DECT imaging of instrumented spines	Dual-Source CT System (SOMATOM Definition Flash, Siemens Healthcare)	The combination of VMI and iMAR demonstrated stronger metal-artifact reduction compared to either algorithm individually
Choo et al., 2021 [88]	Evaluation of the use of VMI and iMAR in DECT imaging of total knee arthroplasty	Dual-Source CT System (SOMATOM Drive, Siemens Healthcare)	Non-VMI images combined with iMAR resulted in the strongest artifact reduction
Yoo et al., 2018 [89]	Evaluation of VMI in patients with metallic implants of the distal radius	Dual-Layer CT System (IQon, Phillips Healthcare)	VMI successfully reduced metal artifacts; 110 to 130 keV offered the best image quality and fewest artifacts.

## Data Availability

Not applicable.

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
