# Peer review of "Updates on the Applications of Spectral Computed Tomography for Musculoskeletal Imaging"

_diagnostics, 2024, doi:10.3390/diagnostics14070732_

Round 1

Reviewer 1 Report (Previous Reviewer 2)

Comments and Suggestions for Authors

I have no other concerns on this paper.

Author Response

Thank you so much!

Reviewer 2 Report (New Reviewer)

Comments and Suggestions for Authors

The authors present a comprehensive literature review of spectral CT applications in musculoskeletal disorders. Such reviews are generally sought after for educational purposes and frequently cited. The manuscript may be further improved by adding a paragraph on search queries or article type limitations (in terms of bias) and performing additional proofreading to correct typos.

Comments on the Quality of English Language

Consider performing an additional check with LanguageTool or similar extension.

Author Response

Thank you so much for your helpful comments. We have added in a methods paragraph on search queries based on your suggestion. We also performed another check of the English language used in the paper. 

This manuscript is a resubmission of an earlier submission. The following is a list of the peer review reports and author responses from that submission.

Round 1

Reviewer 1 Report

Comments and Suggestions for Authors

Dear Prof.  Kjaer,

the submitted manuscript is an (pictorial) review article about the possibilities of spectral CT imaging in musculoskeletal imaging.

The topic is relevant and the manuscript structure is good. Nevertheless, there are several relevant points of criticism that should be revised, otherwise the manuscript cannot and must not be accepted Overall for a review more literature references were missing.

Abstract: good

1. Introduction: good

2. Spectral CT: a table about subgroups would improve the section

technical aspects for dual-energy CT, dual-layer CT, photon-counting CT.....

3. Gout. Please add here a headline, that now clinical applications coming

Literature in this section is partly old, section is to short and superficially

4. Inflammatory arthropathy: section is too short. Give a figure for example

5. Bone marrow edema: Relevant topic, please add  a table with results of current publications, e.g. including sensitivity and specificity

6. Acute knee trauma

7. Degenerative Disease

8.osteoporosis 

9. malignant diseases: please made a subdivision into primary malignant bone disease (e.g. multiple myeloma) and other malignant disease (e.g. metastasis)

10. Metall artifacts: this is a very big topic, please add table with add least 8 current publications and their content. Perhaps subheadings in this section would be helpful.

11. Limitations.

Comments:

-Please adapt the references. These should be only at the end of the manuscript. This is a little bit unprofessional.

-Figure 1: gout at tibiofemoral joint is not the typical location. Please ad more figures. eg. big toe joint 

-Figure 4-8 are adapted from an other publication. This is unscientific, although permission was obtained. Please choose a different approach and added own figures.

Overall an extensive overwork is necessary.

Comments on the Quality of English Language

Dear Prof.  Kjaer,

the submitted manuscript is an (pictorial) review article about the possibilities of spectral CT imaging in musculoskeletal imaging.

The topic is relevant and the manuscript structure is good. Nevertheless, there are several relevant points of criticism that should be revised, otherwise the manuscript cannot and must not be accepted Overall for a review more literature references were missing.

Abstract: good

1. Introduction: good

2. Spectral CT: a table about subgroups would improve the section

technical aspects for dual-energy CT, dual-layer CT, photon-counting CT.....

3. Gout. Please add here a headline, that now clinical applications coming

Literature in this section is partly old, section is to short and superficially

4. Inflammatory arthropathy: section is too short. Give a figure for example

5. Bone marrow edema: Relevant topic, please add  a table with results of current publications, e.g. including sensitivity and specificity

6. Acute knee trauma

7. Degenerative Disease

8.osteoporosis 

9. malignant diseases: please made a subdivision into primary malignant bone disease (e.g. multiple myeloma) and other malignant disease (e.g. metastasis)

10. Metall artifacts: this is a very big topic, please add table with add least 8 current publications and their content. Perhaps subheadings in this section would be helpful.

11. Limitations.

Comments:

-Please adapt the references. These should be only at the end of the manuscript. This is a little bit unprofessional.

-Figure 1: gout at tibiofemoral joint is not the typical location. Please ad more figures. eg. big toe joint 

-Figure 4-8 are adapted from an other publication. This is unscientific, although permission was obtained. Please choose a different approach and added own figures.

Overall an extensive overwork is necessary.

Reviewer 2 Report

Comments and Suggestions for Authors

This review article is focused on the clinical applications of Dual Energy CT (DECT) in musculoskeletal imaging. Despite it's fluent structure and fair selection of relevant publication on the topic, it must be emphasized that there are already several reviews on this topic, which is not new in the diagnostic imaging arena. Even though new technologies have been recently introduces, such as photon counting CT (PCCT), also enabling DECT at detector level, there is no more than few mentions on the fact that some authors have recently (and successfully) used this technologies for some MSK imaging (such as, for instance, in regards to metal artifact reduction). Besides this, I have only to notify a technically wrong sentence at lines 60-61: "Dual-layer CT differs from DECT as this technique uses one X-ray source that emits two different energy spectra, thus helping reduce artifact and spectral separation". Indeed, in dual layer, the single X-ray source emits a beam with a fixed quality (with a given energy spectrum depending on kV and filtration); the detectors is designed in such a way that the first layer is more sensitive to the lower energy component and the second (subsequent) layer detects the residual (hardened) beam emerging from the first layer, containing most of the high energy information. Figures 1 and 2 do not report the original source. Overall, the article follows structures and contents very similar to Rajah et al, AJR Am J Roentgenol 2019 Sep;213(3):493-505, with no big new insights (not surprisingly) except for the mention of the work from Fu et al (Acta Biomaterialia 2023, 166, 615-626) on the use of biosyntetic nanosheets as CA for DECT in osteosarcoma.
